# Prediction of Amino Acid Substitutions in ABL1 Protein Leading to Tumor Drug Resistance Based on “Structure-Property” Relationship Classification Models

**DOI:** 10.3390/life13091807

**Published:** 2023-08-24

**Authors:** Svetlana I. Zhuravleva, Anton D. Zadorozhny, Boris V. Shilov, Alexey A. Lagunin

**Affiliations:** 1Department of Bioinformatics, Pirogov Russian National Research Medical University, 117997 Moscow, Russia; zhuravleva_si@mail.ru (S.I.Z.); zadorozhnyy.ad@gmail.com (A.D.Z.); borisshilov@gmail.com (B.V.S.); 2Department of Bioinformatics, Institute of Biomedical Chemistry, 119121 Moscow, Russia

**Keywords:** sequence-structure analysis, drug resistance, molecular fragments, MNA descriptors, MultiPASS, amino acid substitution

## Abstract

Drug resistance to anticancer drugs is a serious complication in patients with cancer. Typically, drug resistance occurs due to amino acid substitutions (AAS) in drug target proteins. The study aimed at developing and validating a new approach to the creation of structure-property relationships (SPR) classification models to predict AASs leading to drug resistance to inhibitors of tyrosine-protein kinase ABL1. The approach was based on the representation of AASs as peptides described in terms of structural formulas. The data on drug-resistant and non-resistant variants of AAS for two isoforms of ABL1 were extracted from the COSMIC database. The given training sets (approximately 700 missense variants) were used for the creation of SPR models in MultiPASS software based on substructural atom-centric multiple neighborhoods of atom (MNA) descriptors for the description of the structural formula of protein fragments and a Bayesian-like algorithm for revealing structure-property relationships. It was found that MNA descriptors of the 6th level and peptides from 11 amino acid residues were the best combination for ABL1 isoform 1 with the prediction accuracy (AUC) of resistance to imatinib (0.897) and dasatinib (0.996). For ABL1 isoform 2 (resistance to imatinib), the best combination was MNA descriptors of the 6th level, peptides form 15 amino acids (AUC value was 0.909). The prediction of possible drug-resistant AASs was made for dbSNP and gnomAD data. The six selected most probable imatinib-resistant AASs were additionally validated by molecular modeling and docking, which confirmed the possibility of resistance for the E334V and T392I variants.

## 1. Introduction

In recent decades, there has been an increase in oncological diseases all over the world. In spite of considerable efforts in the development of new antineoplastic drugs, oncological diseases hold one of the leading positions in the mortality of the population in developed countries. One of the main problems in the treatment of oncological diseases is drug resistance.

Drug resistance is a well-known phenomenon that occurs when drugs cease to be effective in the treatment of a patient. This phenomenon was first considered when bacterial resistance to certain antibiotics was discovered. However, later, it was found that similar mechanisms occur in other cases, oncological disease being among them. Although many types of cancer are initially responsive to chemotherapy, drug resistance can be developed over time due to various mechanisms, such as DNA mutations and metabolic changes that promote drug inhibition and degradation [1]. There are many different mechanisms of drug resistance [2], but most of them are often related to genetic variations leading to the single amino acid substitution (AAS) in drug target proteins. The early discovery of such an AAS is needed to improve the treatment of patients by changing the drug.

There are various bioinformatics methods for predicting variants associated with drug resistance. All of them are divided into two large groups: methods based on the 3D structure of proteins and methods based on the analysis of protein sequences using machine learning methods [3,4,5,6]. The first group includes molecular modeling, calculation of the structure-dependent binding energy, and a fitness model based on the volume of the binding site. These methods were widely used in the past but required high-quality specialists in structural bioinformatics and computational resources and were associated with studies of individual amino acid substitutions. In many cases, such studies were made to reveal the molecular mechanisms of known drug-resistant substitutions. Therefore, Lin and co-authors studied mechanisms of resistance to crizotinib and lorlatinib in ROS1 fusion-positive lung cancer by structural modeling and comparison of wild and mutated 3D structures of ROS1 (ROS proto-oncogene 1, receptor tyrosine kinase) [7]. This study was limited by seven amino acid substitutions. They showed a severe steric clash against the binding of lorlatinib for the L2086F substitution in ROS1. Wu and co-authors used an ensemble of molecular modeling strategies to theoretically uncover the resistance mechanisms of V550E/L and N535D/K mutations in the receptor tyrosine kinase FGFR4 [8]. They showed that drug resistance to these substitutions was related to weakening the binding affinity of ponatinib. The analysis of the molecular basis of drugs binding to wild-type/mutant ABL1 in 3D complexes is one of the main trends in the computational estimation of possible drug-resistant amino acid substitutions. The 3D structural basis of mutation hotspots (gatekeeper, G-loop, αC-helix, and A-loop) associated with 3318 drug-resistant mutations in 538 human kinases was discussed in the review of Kim with co-authors [9]. Hauser and co-authors used docking and free-energy calculations to estimate the drug resistance of 144 AASs to eight FDA-approved drugs acting as ABL1 inhibitors [10]. Liu and co-authors used the docking of drugs into the mutant ABL1 structure and the estimation of its binding free energy together with IC50 values to predict and experimentally validate 12 drug-resistant AASs related to imatinib, nilotinib, dasatinib, bosutinib, and ponatinib [11]. The second group includes methods for predicting drug-resistant variants using rules and classifiers obtained on the basis of a statistical analysis of sequences of resistant and nonresistant proteins [4,12]. Some recent publications used a combination of structural modeling and machine learning methods [12]. For example, Zhou and co-authors created models and the freely available web application SUSPECT-ABL to predict the drug resistance of AASs to eight FDA-approved drugs [13]. They used machine learning based on the generation and evaluation of a set of features capturing structural, geometric, and physicochemical properties of the protein and ligands and changes in Gibbs free energy of protein-ligand binding affinity based on homology modeling of mutant proteins for 144 ABL1 mutations (19 resistant and 125 susceptible) and docking of the drugs [13]. Aldeghi and co-authors evaluated the ability of three distinct computational methods (structure-based, mixed physics- and knowledge-based potentials, and machine learning) to predict the drug resistance of protein mutations in Abl kinase [14]. Despite the fact that approaches to the creation of such methods were proposed more than 20 years ago, there are still no generally accepted methods, and many methods give conflicting results. The identification of new potential drug-resistant AAS remains complicated and is mostly carried out in scientific laboratories. Therefore, there is an urgent need to create new effective methods for predicting resistant variants associated with AAS in the sequence of target proteins.

Here, we introduce a new method for predicting AASs leading to tumor drug resistance based on a combination of bioinformatics and chemoinformatics approaches. It uses the representation of molecular fragments of protein with AAS as a structural formula and the creation of the “structure-property” relationships (SPR) classification models. The same approach was previously successfully applied to predict the sites of post-translational phosphorylation of proteins [15], pathogenic amino acid substitutions in proteins, including ones associated with monogenic hereditary diseases [16,17], as well as the association of CDR3 regions of T-cell receptors with epitopes and MHC alleles [18]. In this study, this method has been applied to create SPR classification models to predict the drug resistance of AASs in two isoforms of the tyrosine protein kinase ABL1 in relation to imatinib and dasatinib.

Tyrosine protein kinase ABL1 is a main drug target for the treatment of chronic myeloid leukemia (CML). CML accounts for approximately 15% of newly diagnosed cases of leukemia in adults [19]. It was shown that the main molecular basis of CML pathogenesis is the translocation between the arms of chromosomes 9 and 22, leading to the creation of chromosome Philadelphia and BCR-ABL1 gene fusion (~95% of CML patients) [20]. BCR-ABL1 gene fusion leads to the synthesis of chimeric autoactivated proteins, inducing cell proliferation and resistance to programmed cell death [21]. Imatinib, the first ABL1 inhibitor, was launched in 2001 and revolutionized CML treatment, significantly decreasing mortality and increasing survival in patients [19]. CML is considered a chronic disease, and patients apply imatinib for a long time. It was shown that 17% of patients revealed drug resistance, and most patients with advanced-phase disease rapidly became resistant to therapy [22]. The rapid appearance of drug resistance to imatinib in the second to third year of therapy may be related to the presence of clones with drug-resistant AASs at ABL1 [23]. All of this points to the need to identify mutations in ABL1 associated with drug resistance in patients with CML.

## 2. Materials and Methods

### 2.1. ABL1 Protein

The sequences of two human ABL1 tyrosine protein kinase isoforms were obtained from the UniProt database (P00519). For each of the isoforms, the data on AASs were obtained from the COSMIC database (November 2022) [24]. The data included the known AASs (their positions and new amino acid residues), both leading to drug resistance and unrelated to drug resistance to imatinib and dasatinib, the drugs currently used for the treatment of chronic myeloid leukemia CML.

### 2.2. Datasets and Data Preparation

The analysis of the data related to ABL1 isoforms in the COSMIC database revealed 660 AASs for isoform 1 and 696 ones for isoform 2. Next, for each ABL1 isoform, sets of peptides of the appropriate length with AAS in the center were created, as shown in Figure 1. Each set was characterized by a specific peptide length ranging from 3 to 31 amino acid residues. Thus, 15 sets with different peptide lengths were obtained for each isoform. It was made because, at the beginning of the study, we did not know what length of peptides would provide the best accuracy for SPR models. Our previous studies on pathogenic AAS prediction showed that there was no universal peptide length, leading to SPR models with the best predictive accuracy for different proteins [16,17].

Then the peptide sequences were converted into their structural formulas in MOL V3000 format (developed by MDL Information Systems Inc. (now BIOVIA, San Diego, CA, USA) and widely used in chemoinformatics studies) and saved as SD (Structure Data) files. SD files contain structural formulas for definite protein fragments, the amino acid substitution position, the coding protein gene name, and the effect of missense variant labels. Using the original Python script based on the RDKit package (https://www.rdkit.org/ (accessed on 10 December 2022)), we generated datasets in SD files. Furthermore, the generated SD files were used as the training sets for MultiPASS. The SD and Excel files with the length of peptides that leads to the best SPR models are represented in Appendix A.

### 2.3. PASS Software and MNA Descriptors

To predict resistance, it is necessary to create structure-property relationship (SPR) classification models. MultiPASS 2022 software was used for this purpose. It is a special version of PASS 2017 (Prediction of Activity Spectra for Substances) software that was early developed to create SPR classification models based on the representation of structural formulas of amino acid sequences [15,16,17,18]. The PASS software has been successfully used for many years [25], and it was also implemented in several web applications for predicting the biological activity of compounds based on their structural formula [17,18,25,26,27,28,29].

MultiPASS uses a unified set of atomocentric substructural MNA descriptors to represent the structural formula of peptides and a modified Naive Bayes classifier to model structure-property relationships [30,31]. The main feature of the MultiPASS version is the ability to vary the level of MNA (Multilevel Neighborhoods of Atoms) descriptors (from 1 to 15) when describing a structural formula.

MNA descriptors are based on the representation of the molecular structure, which includes hydrogen atoms in accordance with the valency and partial charges of atoms and does not indicate the types of bonds [31]. MNA descriptors are generated as a recursively defined sequence:Zero-level MNA descriptor for each atom is the mark *A* of the atom itself;Any next-level MNA descriptor for the atom is the sub-structure notation *A*(*D*_1_*D*_2_…*D_i_*…), where *Di* is the previous-level MNA descriptor for *i*–th immediate neighbor of the atom *A*.

The mark of an atom may include not only the atomic type but also any additional information about the atom. In particular, if the atom is not included in the ring, it is marked by “-”. The neighbor descriptors *D*_1_*D*_2_*...D_i_…* are arranged in a unique lexicographic order. The iterative process of MNA descriptors generation can be continued, covering the first, second, etc. neighborhoods of each atom (Figure 2).

The substances are considered equivalent in MultiPASS if they have the same list of MNA descriptors. Since MNA descriptors do not represent the stereochemical peculiarities of the molecule, substances whose structures differ only stereochemically are formally considered to be equivalent.

The algorithm of MultiPASS is based on the naive Bayes approach with some significant enhancements. For each class *A_k_* (in this study, the relation of amino acid substitution with drug resistance), which can be predicted by MultiPASS, on the basis of a structural formula represented by the list of MNA descriptors {*D*_1_, …, *D_m_*}, the following values are calculated:*S_0__k_* = 2*P(A_k_*) − 1,
*S_k_* = *Sin*[∑*_i_*
*ArcSin*(*2P(A_k_|D_i_*) − 1)/*m*],
*B_k_* = (*S_k_* − *S_0k_*)/(1 − *S_k_S_0k_*),
where *P(A_k_)* is a priori probability to find AAS with class *A_k_*; *P(A_k_|D_i_)* is a conditional probability of class *A_k_* if the descriptor *D_i_* is present in a set of molecule’s MNA descriptors; *m* is a number of MNA descriptors in the molecule under prediction.

The simplest frequency estimations of probabilities *P*(*A_k_*) and *P*(*A_k_|D_i_*) are given by:*P(A_k_)* = *N_k_/N*, *P(A_k_|D_i_)* = *N_ik_/N_i_*.
where *N* is the total number of structures in the training set; *N_k_* is the number of compounds with class *A_k_*; *N_i_* is the number of compounds containing the MNA descriptor *D_i_* in the structure description; *N_ik_* is the number of compounds containing both the class *A_k_* and the descriptor *D_i_*.

The prediction results of the created SPR models include *Pa* and *Pi* values of AAS for drug resistance to the corresponding drug. *Pa* value means that the query AAS belongs to the class of drug-resistant AASs. *Pi* value means that an AAS belongs to the class of non-drug-resistant AASs. If the user chooses a higher value of *Pa* as a cut-off for the selection of probable drug resistance, the chance to confirm the predicted result by the experiment is also high, but many existing drug-resistant AASs will be lost. For instance, if *Pa* > 0.5 is used as a threshold, about half of real drug-resistant AASs will be lost; for *Pa* > 0.7, the portion of lost drug-resistant AASs is 70%, etc. We believe that any case with a *Pa* > *Pi* value needs to be regarded as a positive drug-resistant prediction. By definition, the probability *Pa* and *Pi* values are measures of belonging to both subsets of “drug-resistant AASs” and “non-drug-resistant AASs”.

By definition, the probabilities *Pa* and *Pi* are measures of belonging to both subsets of “active” and “inactive” molecules and the probabilities of the 1st and 2nd types of prediction error, respectively. These two interpretations of the probabilities *Pa* and *Pi* are equivalent and can be used for interpreting the results of a prediction. They can also be used for the creation of different criteria for the analysis of prediction results corresponding to specific tasks.

### 2.4. Approach Estimation

The AUC value (AUC_LOO CV_) based on the leave-one-out cross-validation procedure was calculated during the creation of each SPR model. It was completed automatically in MultiPASS. The best SPR model for the prediction of AASs related to drug resistance was selected based on the highest value of AUC_LOO CV_. For the best SPR models, the 20-fold cross-validation (20-F CV) procedure was performed to calculate the AUC value (AUC_20-F CV_).

### 2.5. Molecular Modeling and Docking Procedure

In the structure modeling part of this work, we used the Robetta.Bakerlab.org web service for amino acid substitution structure modeling of the ABL1 protein. Robetta.Bakerlab.org is a web server that provides computational tools for protein structure prediction, modeling, and design [32]. The given models were used for docking simulations in AutoDock 4.2. AutoDock is specific software for the protein-ligand interaction investigation. It was used jointly with AutoDockTools to prepare proteins and ligands for all steps of the docking procedure [33].

All molecular visualizations were made by the PyMOL software (Schrodinger, LLC., New York, NY, USA, The PyMOL Molecular Graphics System, Version 2.5).

The R-studio server landed on the computational cluster of the Institute of Translational Medicine at RSMU and was used for statistical estimation of results and boxplot plotting [34].

The next list of amino acid substitutions in ABL1 isoform 1 was modeled: A288T, E334V, T392I, F283V, E279Q, and E279G. In this study, a publicly available three-dimensional structure from the Protein Data Bank (PDB) database (PDB ID: 5hu9) was used. The 5hu9 complex was chosen from the PDB as a template for homology modeling for the following reasons:

The structure exhibited high validation scores according to the wwPDB validation data, surpassing the parameters of most similar structures in the PDB database, indicating the quality of the X-ray diffraction investigation.The structure consisted of a single chain, facilitating the preparation process of the three-dimensional structure for docking procedures.The examined structure of the wild type of human ABL1 possessed a high crystallographic resolution of 1.53 Å ensuring enhanced precision during the docking simulations.The presence of the CHMFL-074 inhibitor within the model was situated in the same region of the protein as in other examples of ABL1 protein structures from the PDB, where the inhibitor imatinib was located. This enabled us to utilize the ligand coordinates of CHMFL-074 for modeling the docking of ABL1 with imatinib.”

After successful modeling at Robetta.Bakerlab.org, all model files were downloaded in the PDB format.

Docking simulations using AutoDock included several steps. Firstly, all models were spatially aligned, and binding site coordinates for imatinib were detected. Then all models and imatinib were prepared for the docking procedure. The step includes the removal of all heteroatoms and solvent molecules. Furthermore, each PDB file was converted to PDBQT format, as described in the AutoDock manual.

In our study, a flexible docking protocol was employed, which allowed for the variation of side-chain conformations in the ligand-binding region of the protein. Consequently, the protein’s flexibility in the binding site was considered. As the main objective of the investigation was to highlight the differences in binding parameters with imatinib for various protein mutations, it was decided to limit the study to docking procedures and forego molecular dynamics simulations. Liu and co-authors also used only docking without molecular dynamics simulations in their study to estimate the difference in the interaction of drugs with wild and mutant ABL1 [11].

The results of the docking are based on binding energy estimations. The binding energy calculation in AutoDock involves the summation of several types of energy terms, including electrostatics, van der Waals interactions, hydrogen bonding, and others. These energy terms are calculated based on the positions and charges of the atoms in the ligand and receptor molecules, and they are combined to yield an overall estimate of the binding affinity between the two molecules.

Independent assessments for 100 conformations of the ligand in the ABL1 binding site were made for each model, and then the statistical approach for the estimation of success in docking was used. We also got those assessments for control docking to estimate interactions between imatinib and ABL1 without any amino acid substitutions (wild type).

## 3. Results

### 3.1. Selection of the Best SPR Models

Based on COSMIC data, 660 variants of AASs were found for ABL1 isoform 1, of which 84 AASs were associated with the resistance to imatinib and 3 AASs were associated with the resistance to dasatinib. For the second isoform щa ABL1, 696 AASs were identified, of which 30 AASs were associated with resistance to imatinib. The remaining AASs were considered not to be associated with drug resistance.

The influence of the level of MNA descriptors and the size of peptides with centered AAS in the training set on the accuracy of SPR models was investigated. SPR models were built based on the training sets with various peptide lengths (from 3 to 31 amino acid residues) and different levels of MNA descriptors (from 1 to 15). As a result, 225 SPR models (15 training sets with different lengths of peptides on 15 levels of MNA descriptors) were created for each isoform of the ABL1 protein. It was revealed that the 6th level of MNA descriptors and the peptide length of 11 amino acid residues were the best combination for ABL1 isoform 1 that achieved the highest average AUC value (0.915) of SPR models (Figure 3A). The highest prediction accuracy of AASs related to drug resistance to imatinib (0.908) for ABL1 isoform 2 was revealed for the combination of the 6th level of MNA descriptors and the peptide length of 15 amino acid residues (Figure 3B).

Table 1 shows the numbers of resistant and nonresistant AASs for the appropriate drug in the training sets and the AUC values of the best SPR models for ABL1 isoforms. It displays the high accuracy of the prediction of AASs related to drug resistance. At the same time, the accuracy of the prediction of drug-resistant AASs against imatinib for the second isoform is better (0.909) than for the first one (0.851). The accuracy of prediction for AASs related to dasatinib resistance is very high (0.980). The validation of the best SPR models shows that the AUC_LOO CV_ and AUC_20-F CV_ values are close to each other. It means that the given SPR models are considered robust and may be used for the estimation of possible drug-resistant AASs.

### 3.2. Comparison Prediction Results of MultiPASS and SUSPECT-ABL

There are several publications related to the creation of methods for predicting drug-resistance AASs in ABL1 [10,11,12,13]. Unfortunately, the sets of AASs related to imatinib or dasatinib drug resistance used in these studies were too small to make a direct statistically significant comparison with the MultiPASS approach. Nevertheless, the recently developed, freely available web service SUSPECT-ABL [13] provides a possibility for such a comparison. SUSPECT-ABL has a fairly convenient and intuitive interface that allows users to simultaneously predict the drug resistance of up to 50 AASs to eight FDA-approved ABL1 inhibitors. SUSPECT-ABL makes predictions of drug resistance for AASs within the kinase domain (from position 242 to 493). In this interval, there were 271 AASs in the training set, including 82 associated with drug resistance to imatinib. Due to the small number of AASs related to dasatinib resistance, in this comparison only imatinib resistant AASs were used. All 271 AASs were evaluated for drug resistance against imatinib by SUSPECT-ABL and MultiPASS (the best SPR model for ABL1 isoform 1 mentioned in Table 1 was used). For each AAS, its data were excluded from the training set and the SPR model during the prediction of imatinib-resistance by MultiPASS, so that this variant was perceived by the model as a new one. The comparison of predicted results of MultiPASS and SUSPECT-ABL is represented in Table 2 and in Appendix A.

Table 2 shows that SUSPECT-ABL predicted a rather small number of drug-resistant variants; however, over 40% of them were correct (5 TP of 12 (TP + FP)). At that time, the majority of known AASs related to drug resistance to imatinib (77 FN) were not predicted. SUSPECT-ABL demonstrates a high level of specificity (0.958) but a low level of sensitivity (0.061) and a lower balance accuracy (0.509) in comparison with the results given by the MultiPASS SPR model. It may be due to the small number of imatinib-resistant and imatinib-non-resistant (susceptible) AASs used for the creation of SUSPECT-ABL models (5 and 16, respectively) which were taken from the publication of Hauser with co-authors [10].

In contrast to SUSPECT-ABL, the prediction results of MultiPASS may be analyzed using a different threshold of Pa values. The values of TP (78), FP (102), and sensitivity (0.951) are the highest, and the values of TN (87), FN (4), and specificity (0.460) are the lowest by default threshold (Pa > Pi). It means that almost all imatinib resistant AASs in the set were predicted correctly, but more than half of the susceptible AASs were also predicted to be drug-resistant. At that point, the value of balance accuracy for MultiPASS (0.706) was significantly higher compared with that given by the results of SUSPECT-ABL (0.509). By increasing the Pa value as a threshold, we change the ratio of specificity and sensitivity values. Table 2 shows that Pa > 0.5 is the optimal threshold with the highest value of balance accuracy (0.822). Moreover, the majority of imatinib-resistant AASs were predicted correctly ((71/82) × 100 = 86.6%), and only ((42/189) × 100 = 22.2%) susceptible AASs were predicted as drug-resistant. Therefore, this threshold was used for the estimation of drug-resistant AASs from the gnomAD and dbSNP databases (see below). The decreasing number of false positives with the increasing Pa value in Table 2 shows that the chance of discovering a real relationship between AAS and drug resistance significantly increases for the AASs with the highest Pa values. For example, 24 (80%) of 30 predicted drug-resistant AASs at the threshold Pa > 0.9 are drug-resistant, according to the COSMIC database.

### 3.3. Estimation of Drug-Resistant AASs from gnomAD and dbSNP Databases

It is known that when using targeted anticancer drugs, even in the presence of known biomarkers, a positive response is not always observed in patients. We applied the created SPR models of ABL1 isoform 1 for the estimation of drug-resistant AASs from the gnomAD (249 AASs) and NCBI dbSNP (508 AASs) databases, which were different from the AASs of the training set. The number of AASs for ABL1 isoform 2 was too small to make such a study. The prediction results are provided in Appendix A as SD and Excel files. The prediction results for gnomAD data showed that 13 AASs were predicted with a probability of Pa > Pi values as resistant to imatinib, but there was no predicted AAS with a Pa value greater than 0.5. There were 31 and 9 AASs from dbSNP with the probability of being resistant to imatinib and dasatinib, respectively. Of them, only six AASs were predicted to be resistant to imatinib, with Pa values ranging from 0.575 to 0.743 (Table 3).

The six selected potential drug-resistant AASs were analyzed by molecular modeling and docking (as described in Section 2) to estimate how they may influence on the interaction between ABL1 and imatinib.

For the native protein, the median value of the binding energy of the ABL1-imatinib complex per 100 conformations was calculated; the value of the indicator was −10.590 kcal/mol. The values of the binding energy of the ABL1-imatinib complex varied from −11.275 to −10.015 kcal/mol. The comparative analysis of the binding energy of mutant proteins and wild-type proteins showed a statistically significant increase in binding energy for two structural models: the mutations E334V and T392I (Figure 4). The median value of the binding energy of the protein with E334V AAS was −9.770 kcal/mol (the binding energy varied from −10.265 to −9.315 kcal/mol), which is 1.08 times higher (*p* = 9.007 × 10^−11^) than that of the protein without mutant changes. The median value of the binding energy of protein with T392I AAS was −10.32 kcal/mol (the binding energy varied from −10.77 to −9.47 kcal/mol). This value also turned out to be significantly higher than that of the wild-type protein (*p* = 0.0015).

PyMOL visualization of structural models with the interactions of imatinib with the wild-type protein and protein with E334V and T392I variants was considered an example of studying the effect of AAS on the protein-ligand complex (Figure 5). It showed the difference in the number of hydrogen bonds (the protein-ligand complex with wild-type ABL1 isoform 1 had two hydrogen bonds while the complex with mutated ABL1 isoform 1 had one hydrogen bond) and the change in the distance between the interacting atoms (for ABL1 isoform 1 with E334V AAS, the distance between the interacting atoms was increased from 2.1 A to 2.6 A). Therefore, E334V and T392I variants can explain the reduction in the protein binding strength of imatinib and a decline in its pharmacotherapeutic effectiveness. At that point, it should be mentioned that both variants (E334V and T392I) are far from the binding site of imatinib, and they change the conformation of the protein in such a way that they disrupt the interaction with the inhibitor. On the other hand, T392I is in the A-loop (the activation loop including amino acid residues 381–402) of ABL1 [35], which is considered an important region for drug-resistant mutations related to imatinib. Imatinib binding to the inactive state of the kinase domain with the activation loop in the “closed” conformation and the changing A-loop conformation may influence this process [36]. E334V variant is in turn 332–334 according to UniProt data related to 5hu9. Our study revealed that the conformational change in this region may be related in imatinib resistance.

According to UniProt, data related to the 5hu9, E279G, and E279Q variants are in between the turn in positions 275–277 and the helix in positions 280–290; at the same time, the F283V and A288T variants are in the helix in positions 280–290. Although the other four predicted AASs did not show impairment of ABL1 binding to imatinib in molecular modeling and docking, it is possible that their combination could lead to such changes at compound mutations. However, this requires further research.

## 4. Discussion

The application of New Generation Sequencing (NGS) technology has become a necessity in developing personalized treatment for patients with tumor diseases. The use of NGS technology for reviling mutations related to drug resistance is also important. It is confirmed by the 2020 European LeukemiaNet recommendations, which include detection of BCR-ABL1 mutations [37]. The massive use of NGS technology for patients with tumor diseases will lead to the appearance of new variants of amino acid substitutions, which should be annotated from a drug resistance point of view. Therefore, the new high-performance and accurate computational tools for the prediction of drug-resistant variants are highly actual.

At the present time, there are dozens of targeted antineoplastic agents that are effectively used in treatment. Unfortunately, drug resistance appears in a significant number of patients. Over the past decade, a significant amount of data related to genomic variants leading to AASs in tumors has been revealed and collected at the COSMIC database. Some of them are known to cause resistance against the appropriate drugs. The availability of such data provides bioinformaticians with the possibility of developing approaches to predict AASs related to drug resistance. Our study displays a successful combination of bioinformatics and chemoinformatics methods (sequence-structure analysis), leading to the creation of high-performance and accurate structure-property relationship classification models for the prediction of AASs related to individual resistance to antineoplastic drugs. As a practical application of these models, they have been applied to AASs in the ABL1 protein from the gnomAD and NCBI dbSNP databases to reveal new possible variants of drug resistance. As a result, using the created SPR model, six potential drug-resistant AASs have been predicted with a high probability. Based on molecular modeling and docking, two of them (E334V and T392I) have also been established as potential disruptors of protein-ligand interactions. We hope that this finding will attract the attention of oncologists.

The described approach is based only on information about amino acid sequences of proteins and drug-resistant or susceptible AASs. It has some advantages as it does not require high-resolution 3D structures of protein-ligand complexes that are not always available. Moreover, due to technical difficulties, only a part of the proteins in such 3D complexes is quite often obtained, and this does not allow the analysis of drug-resistant mutations along the entire length of the protein sequence. In our study, we wanted to demonstrate how well this approach works without involving additional characteristics or data related to protein features. At the same time, it is known that some drug-resistant AASs are located in the secondary structures of proteins, such as helices, loops, and strands, and may lead to disruption of their structure. For example, Kim and co-authors [9] systematically investigated four types of representative mutation hotspots (gatekeeper, G-loop, αC-helix, and A-loop) associated with drug resistance in 538 human protein kinases. They described AASs E255K/V and Y257C in ABL1 as the representative G-loop mutations that disrupt an electrostatic triad required for imatinib-binding, causing cell resistance to imatinib [38]. Zhang and co-authors used the analysis of AASs within the β3-αC loop of EGFR/ERBB2/BRAF/MAP2K1 to predict responses to therapies [39]. Zhou and co-authors also highlighted the importance of the AASs influence on the conformations of the ABL1 phosphate-binding loop (P-loop) and the activation loop (A-loop) [13]. Therefore, despite the fact that the peptides used in the creation of models can indirectly reflect the features of the secondary structure of the protein in the corresponding positions of AASs, the use of data on the secondary structure of the protein as well as other features of the protein (for example, a domain or an interdomain region, an active site or a center of allosteric regulation, etc.) in the sites of AASs as additional variables may improve the accuracy of predicting the relationship between AASs and drug resistance.

In the future, we plan to study the influence of additional above-mentioned features of proteins on the accuracy of SPR models. We also plan to create SPR models to predict drug resistance for AASs in other proteins that are drug targets in antineoplastic therapy and create a freely available web application that will help clinicians find the possible reasons for drug resistance related to not-annotated AASs.

## Figures and Tables

**Figure 1 life-13-01807-f001:**
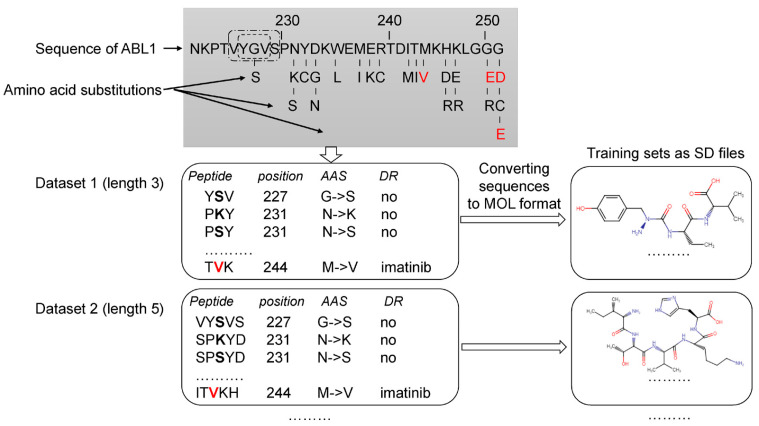
Creation of the training sets. Drug-resistant and nonresistant AASs were revealed in COSMIC and annotated. The AASs were mapped on ABL1 sequence. A peptide with the appropriate length and AAS in the center was generated for each AAS position. According to the length of peptides (from 3 to 31 amino acid residues), 15 datasets were created. These datasets were converted to 15 SD files with description of structural formulas of peptides in MOL V3000 format and classes of annotation. Red letters—AAS associated with drug resistance to imatinib. Black bold letters—AASs not associated with drug resistance. DR—drug resistance.

**Figure 2 life-13-01807-f002:**
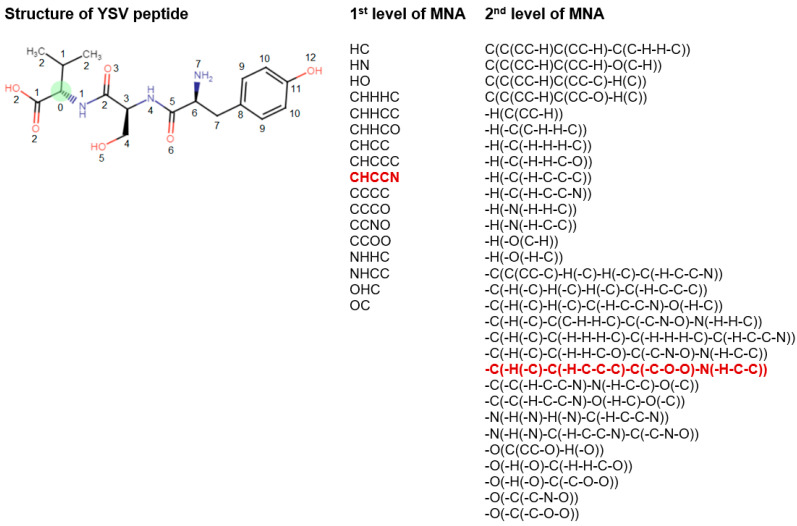
Description of structural formula of YSV peptide by the lists of 1st and 2nd levels of MNA descriptors. The numbers close to the atoms display the most distant atoms included in the MNA descriptor of the appropriate level for green C atom. The selected MNA level is used in the model, sets of such descriptors are generated for the each of atoms. Red—MNA descriptors of 1st and 2nd levels for green C atom. The name of atoms is considered the zero level of MNA descriptors.

**Figure 3 life-13-01807-f003:**
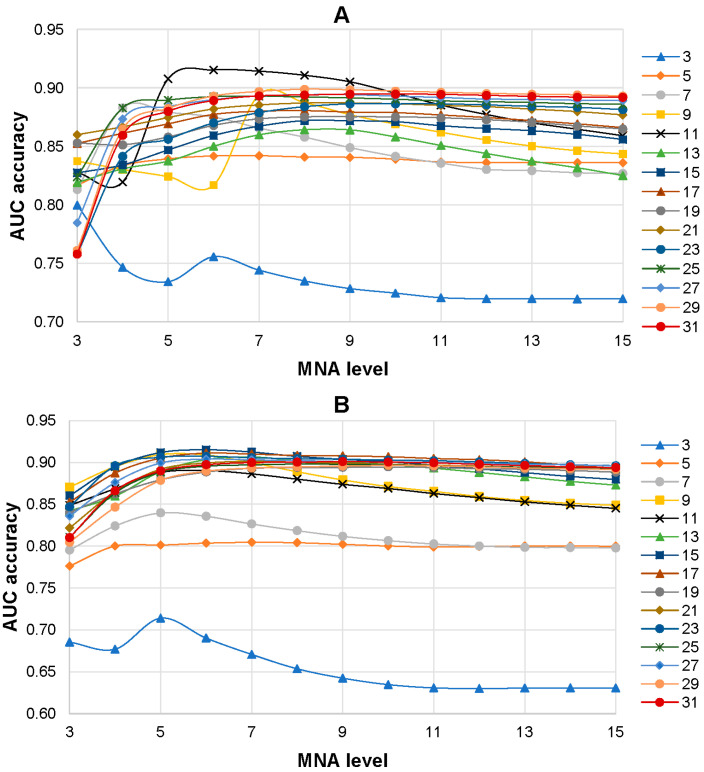
Dependence average AUC values from the level of MNA descriptors and the length of peptides in the training set for ABL1 1st (**A**) and 2nd (**B**) isoforms. The legend of the figure shows the correspondence between the color with signs and length of the peptides in the training set.

**Figure 4 life-13-01807-f004:**
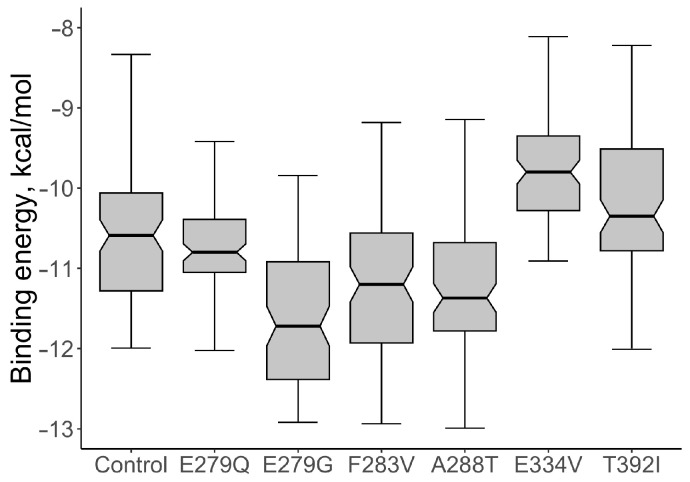
The comparative analysis of the binding energy values of ABL1-imatinib complexes with selected AASs and wild type of ABL1 isoform 1 (Control).

**Figure 5 life-13-01807-f005:**
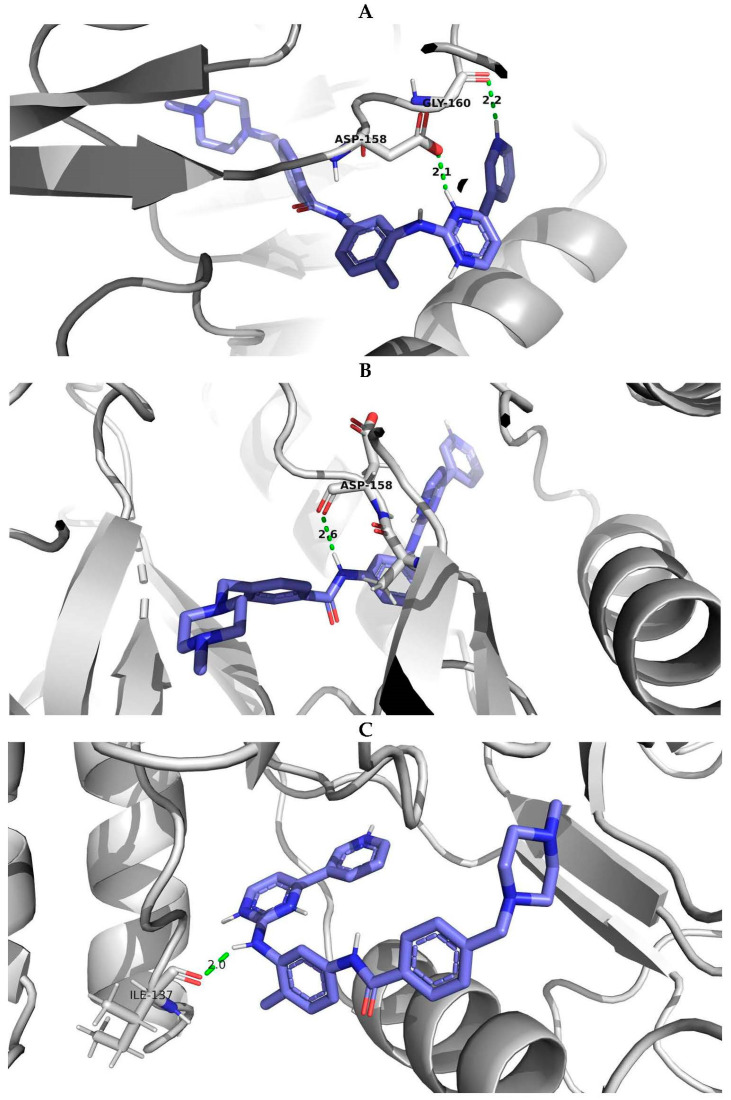
PyMOL visualization of imatinib-ABL1 interaction for (**A**) wild type of protein, (**B**) protein with E334V amino acid substitution and (**C**) protein with T392I amino acid substitution. Blue molecule—imatinib; Green—hydrogen bonds; Red—oxigen; Dark blue—nitrogen.

**Table 1 life-13-01807-t001:** The data of SPR models for ABL1, isoform 1,2 at the 6th level of MNA descriptors and a peptide length of 11 (*ABL1*, *isoform 1*) and 15 (*ABL1*, *isoform 2*).

Number of Drug Resistance AASs	Number of Not Drug Resistance AASs	Drug Resistance to	AUC_LOO CV_	AUC_20-F CV_
*ABL1*, *isoform 1*				
84	576	imatinib	0.851	0.846
3	657	dasatinib	0.980	0.980
*ABL1*, *isoform 2*				
30	670	imatinib	0.909	0.907

**Table 2 life-13-01807-t002:** Comparison between prediction results of MultiPASS and SUSPECT-ABL.

Software	TP	FP	TN	FN	Sen.	Sp.	Bal. Acc.
SUSPECT-ABL	5	8	181	77	0.061	0.958	0.509
MultiPASS, Pa > Pi	78	102	87	4	0.951	0.460	0.706
MultiPASS, Pa > 0.3	77	87	102	5	0.939	0.540	0.739
MultiPASS, Pa > 0.4	76	64	125	6	0.927	0.661	0.794
MultiPASS, Pa > 0.5	71	42	147	11	0.866	0.778	0.822
MultiPASS, Pa > 0.6	66	32	157	16	0.805	0.831	0.818
MultiPASS, Pa > 0.7	60	26	163	22	0.732	0.862	0.797
MultiPASS, Pa > 0.8	42	18	171	40	0.512	0.905	0.708
MultiPASS, Pa > 0.9	24	6	183	58	0.293	0.968	0.630

TP—True Positives; FP—False Positives; TN—True Negatives; FN—False Negatives; Sen.—Sensitivity = TP/(TP + FN); Sp.—Specificity = TN/(TN + FP); Bal. Acc.—Balance Accuracy = (Sen. + Sp.)/2.

**Table 3 life-13-01807-t003:** AASs predicted to be resistant to imatinib with the probability more than 0.5.

AAS	Number of MNA Descriptors	Number of New MNA Descriptors	Pa	Pi	Drug Resistance
E279G	839	21	0.743	0.042	Imatinib
E279Q	778	11	0.742	0.043	Imatinib
T392I	1002	17	0.671	0.055	Imatinib
F283V	635	1	0.671	0.055	Imatinib
E334V	693	1	0.603	0.064	Imatinib
A288T	674	10	0.575	0.068	Imatinib

Number of MNA descriptors—number of MNA descriptors describing the structure of the peptide with AAS that are in the training set; Number of new MNA descriptors—number of MNA descriptors describing the structure of the peptide with AAS that are not in the training set. It may be used for the estimation of the applicability domain. The higher number of new MNA descriptors, the more different is the peptide structure from those from the training set.

## Data Availability

All data on AASs from COSMIC, gnomAD (http://gnomad-sg.org/ (accessed on 10 April 2023)) and NCBI dbSNP (https://www.ncbi.nlm.nih.gov/snp/ (accessed on 10 April 2023)) databases are provided in SD and Excel files in Appendix A. The data on resistant and nonresistant ABL1 amino acid substitutions used for the creation of training sets were given from COSMIC database (https://cancer.sanger.ac.uk/cosmic (accessed on 10 November 2022)).

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
