# Peer review of "Prediction of Amino Acid Substitutions in ABL1 Protein Leading to Tumor Drug Resistance Based on “Structure-Property” Relationship Classification Models"

_life, 2023, doi:10.3390/life13091807_

Round 1

Reviewer 1 Report

Here, the authors aim at prioritizing potential imatinib/dasatinib resistance-related amino acid substitutions on the non-receptor tyrosine-protein kinase ABL1 through implementing structure-property relationship classification models. Although such computational approaches are promising and still underexploited tools for the prediction of drug-resistant mutants of kinases and other oncoproteins, the present study has some flaws that undermine its scientific merit.

1) While dissecting ABL1 into smaller peptides for SPR classification studies, it could also be useful to create structurally relevant fragments such as helices, loops and strands. Please discuss this in the manuscript and provide examples of how similar investigations are conducted in the relevant scientific literature.

2) There is no scientific rationale behind target selection for in silico mutagenesis and molecular docking studies. In the PDB, there also exist 3D structures of human ABL1 kinase domain in complex with dasatinib (e.g. 2GQG) or imatinib (e.g. 2HYY). Please provide a detailed explanation for this.

3) The authors did not allow flexibility for the protein during docking. ABL1, however, is known to have multiple conformational states. Furthermore, the specific conformations of the P-loop, activation loop and hinge region of the enzyme are also believed to allow for productive drug binding. Please support the validity of the predicted protein-ligand complexes with additional molecular dynamic simulations. Also, please refer to the scientific literature on the molecular basis of dasatinib/imatinib binding to wild-type/mutant ABL1.

Minor editing of English language required.

Reviewer 2 Report

This study developed a classification model which can predict the influence of single amino acid substitution (AAS) for ABL1 protein. This model can be used to analyze tumor-drug resistance by providing the binding affinity change caused by AAS.

One main issue of this manuscript is the evaluation part, which only showed the “internal” accuracy comparison between different model settings (Fig. 3). But more importantly, an “external” comparison should be shown between this model and other work. As mentioned in the Introduction, “There are various bioinformatics methods for predicting variants associated with 56 drug resistance.” It is of great interest for readers and potential users to know if this work is significantly better than previous work.

Secondly, the data represented in Fig. 3 is hard to distinguish, since the curves are overlapped with each other. Also, the colors for annotating different length of peptides are too similar, e.g., there are 4 different blues.

Minor issue: there are some wrong sentences or typos that need to be fixed.

Line 14: “more 0.9” should be “more than 0.9”

Lines 106 -107: “It was shown that at 17% of patients revealed drug resistance and most patients with advanced-phase disease rapidly became resistant to therapy.” “at 17% of patients” should be “17% of patients”.

Line 142: “ASSs” should be “AASs”.

Reviewer 3 Report

The manuscript is well-conceived and developed. Only minor concerns about molecular docking studies, which could be discussed more deeply in the experimental section and in results. 

Figure 5 should be revised being poorly informative.

Round 2

Reviewer 1 Report

The authors have now improved the quality of their manuscript significantly.